# Sustainability Reports and Disclosure of the Sustainable Development Goals (SDGs): Evidence from Indonesian Listed Companies

**Herenia Gutiérrez-Ponce** [1,*] and **Sigit Arie Wibowo** [1,2]

1    Department of Accounting, Faculty of Economics and Business, Universidad Autónoma de Madrid, 28049 Madrid, Spain; sigit.wibowo@estudiante.uam.es
2    Department of Accounting, Faculty of Economics and Business, Universitas Muhammadiyah Yogyakarta, Yogyakarta 55183, Indonesia
*    Correspondence: herenia.gutierrez@uam.es

**Abstract:** This study investigates the factors that determine disclosure of the Sustainable Development Goals (SDGs) of companies listed on the Indonesian stock exchange in the period from 2017 to 2021. The research was conducted through an exploratory study using panel data (from each company's websites), parametric correlations, and regression models. The findings show a 60% increase in the disclosure of the SDGs in sustainability reports from 2017 to 2021, with the highest level of disclosure achieved for SDG 3 (Health and well-being) and SDG 4 (Quality education). The lowest disclosure was for SDG 14 (Life below water). The study demonstrates statistically that governance factors such as the presence of women on the board of directors and the number of board meetings positively affect SDG disclosure in listed companies in Indonesia. Factors related to companies' profitability, environmental sensitivity, and board size do not, however, influence SDG disclosure. These findings have implications for academics, stakeholders, practitioners, and governments who are strategically positioned to achieve the SDG agenda in 2030. This study has limitations in that the data were drawn only from companies in the SRI-KEHATI Index.

**Keywords:** sustainable development goals; sustainability reports; SDG reporting; SRI-KEHATI Index; governance level; company characteristics

## 1. Introduction

The SDGs were established in 2015 by the United Nations General Assembly (UN-GA). They consist of 17 goals and 169 targets and are intended to be achieved by 2030 to reduce inequality between nations [1–3]. Organizations, companies, and regulatory authorities are working on development and dissemination of non-financial information through publication of sustainability reports, sustainable investments, green banking, environmental issues, human rights, climate change, etc. All of these topics are also topics of study from both a theoretical and an academic perspective [4–9].

In general, a country's level of development and well-being depends on its internal factors and regional environment [10]. Government policies are thus a fundamental element for implementation of the SDG agenda, but achieving effective implementation requires the collaboration of companies and institutions [11,12]. Companies must ensure that they do not hinder achievement of the SDGs [13], since achievement of the SDGs also affects the organization's long-term sustainability [14,15].

The SDGs are a universal call to action to end poverty, protect the planet, and improve the lives and prospects of people worldwide. In 2017, Indonesia issued Regulation No. 59, which contained a commitment to support achievement of the SDGs by 2030. Article 1 details the parties other than the central government involved in this effort. In the same year, the Indonesian government also issued Financial Services Authority Regulation No.

51/POJK.03/2017, which regulates implementation of sustainable finance for financial services institutions, issuers, and listed companies. This regulation requires Indonesian companies to play an important role in corporate sustainability by publishing annual sustainability reports [16].

Also in 2017, at the Development Goals Summit (DGS), the Indonesian government declared that progress toward achieving the SDGs at regional level is scarce, given the high development gap. The government promised to reduce the development gap and strengthen implementation of the SDGs at the local level. In developing countries such as Indonesia, therefore, organizations must pay attention to the four pillars of sustainable development—economic, social, environmental, and institutional—to achieve the goals for SDG implementation by 2030.

Building on the above-mentioned premises, this study aims to fill an important research gap by analyzing the relationship between disclosure of SDGs and the factors that characterize companies listed on the Indonesian stock exchange in the years following the obligation to publish the sustainability report.

Stakeholder theory suggests that profitable companies work harder to preserve their positive reputation and need to continually address their stakeholders' expectations. Sustainability activities show the efforts companies make to satisfy stakeholders' demands [17,18]. Legitimacy theory also suggests that profitable companies undertake more sustainability activities to manifest their contribution to society's well-being and legitimize their existence [19]. Disclosure of progress toward SDGs is also very important to their stakeholders because it is an additional investment and shows legitimacy, bolstering a company's good reputation [20]. Companies that actively participate in achieving the SDGs experience obstacles, however, which are related to factors beyond their internal control and the value chain as a whole [21,22].

Previous research has found a relationship between some characteristics of companies —size, income, gender diversity on the board of directors (BOD), among others—and the dissemination of specific SDGs [2,23–25]. Marketing companies [26] and multinational companies are also promoting SDG disclosure [27].

Other studies have shown that SDG disclosure is still only symbolic and that corporate engagement in implementing them is still limited and more intentional than actual. These studies also reveal that the measurement of level of corporate engagement in the SDGs is associated with various methodological difficulties, related mainly to selection of indicators, data availability, and interpretation [22,28]. Lack of commitment to the SDGs can sometimes be explained by lack of knowledge, lack of understanding of SDG practices, and other sociological and economic factors in different regions [22,29]. The dissemination of the SDGs thus continues to raise intense academic debate about what factors explain or trigger their dissemination by companies and organizations [2,30].

Based on previous studies, and to achieve our study objective, which focuses on listed companies in a developing country, we pose the following research questions: RQ1. What level of SDG disclosure do companies listed on the Indonesian stock exchange report? RQ2. What SDGs are disclosed most often and prioritized by stock market companies in Indonesia? RQ3. What factors determine SDG disclosure among Indonesian companies?

We analyzed the factors that determine disclosure of the SDGs of companies listed on the Indonesian stock exchange in the period from 2017 to 2021. The research was performed through exploratory study using panel data (from each company's websites), parametric correlations, and regression models.

The dependent variable is the level of SDG disclosure, calculated from total SDG disclosure divided by the 17 SDG targets. The data were obtained from financial and sustainability reports published on the companies' websites. The independent variables analyzed were financial profitability (ROE), environmental sensitivity of the industry or sector, size of the administrative board, gender diversity on the board, and number of board meetings. In addition to addressing the specific research gap (level of disclosure of the SDGs and prioritization of each of the 17 SDGs), we examine the relationship of the

SDG level and the factors that determine SDG disclosure to the variables analyzed, using statistical correlations. Regression models constitute the research methodology.

To the best of our knowledge, this is one of the few studies to examine the level of SDG implementation, and its relationship to the factors that determine this achievement, behind Indonesian listed companies' obligation to submit sustainability reports.

This study makes several contributions to the literature on SDG development in Indonesia and the factors that determine companies' implementation, as each country must evaluate progress in its implementation of the United Nations 2030 Agenda, involving governments and stakeholders [31–33]. Our study specifically shows a 60% increase in disclosure of the SDGs in sustainability reports between 2017 and 2021. The highest disclosure levels are achieved for SDG 3 (Health and well-being) and SDG 4 (Quality Education). The lowest disclosure is for SDG 14 (Life below water). Statistically, governance factors such as presence of women on the BOD and number of board meetings have been shown to have a positive effect on SDG disclosure in listed companies in Indonesia. Factors related to profitability, companies' environmental sensitivity, and BOS do not, however, influence SDG disclosure.

The results of this study are also useful for monitoring companies' sustainability performance after issuance of the 2017 Regulation No. 51, and for determining what specific factors' influence can be useful for decision makers such as managers and investors [34,35]. This study examines specifically where Indonesian companies prioritize their SDG activities as a government strategy to support reporting, as well as a form of commitment to achieving the SDGs themselves [36].

This study is divided into the following sections: Section 1: Introduction; Section 2: Theoretical background, literature review, and hypothesis development; Section 3: Research methodology; Section 4: Results and discussion; and Section 5: Conclusions, implications, research limitations, and future research opportunities.

## 2. Theoretical Background and Literature Review

Some theories used to analyze companies that conduct sustainability activities are agency theory, interest group theory, and legitimacy theory. Jensen and Meckling [37] adopt agency theory when a difference of interests exists between managers (agents) and owners (directors). Whereas managers focus their goals on generating maximum profit to obtain bonuses or compensation for good performance, owners and shareholders want to obtain high profitability without incurring excessive managerial costs. Agency theory suggests that companies adopt more sustainable practices in periods of low profitability as a means of convincing financial actors that the current sustainable initiatives will result in long-term growth in results and competitive advantage for the company [38].

Freeman et al. [39] define stakeholder theory as the view that organizations are composed of a set of actors, which it calls interest groups (stakeholders), and that the purpose of a business is to create the greatest possible value so that the interest groups can be successful and sustainable over time. Both theories relate to SDG activities and encourage companies not only to focus on company profitability but also to protect, develop, and improve community welfare in the company's environment. This is the background against which the company should align its corporate activities with the SDG agenda.

The SDG framework can expand the way companies present their non-financial reporting [40]. Yet the impact of SDG reporting is difficult to assess, as creating value for shareholders is not always easy to quantify [41]. Some companies believe it is not important to report on their non-financial activities [42]. Integration of non-financial or sustainability and financial information has, however, become a new paradigm among researchers [43–45].

Aligning the SDG goals with value creation for shareholders becomes difficult on its own [30], and the absence of reporting standards makes it hard for companies to convey information on their achievement of SDGs. The United Nations and WBCSD Global Reporting Initiative, United Nations [46] have sought to solve this problem by introducing

an SDG Compass to permit companies to measure and see their contribution while aligning corporate strategies to achieve SDGs.

The SDG Compass explains how SDGs affect a company's business by providing the tools and insights needed to put sustainability at the center of one's business strategy. According to PWC studies 2018 [40] and 2019 [47], companies have taken important steps in disclosing the SDGs from year to year. According to Emma and Jennifer [6], SDG disclosure influences companies' reputation, competitive advantage, and financial performance. Institutional investors also believe that SDG disclosure is the best way to evaluate environmental and social performance [23].

According to the Secretary General of the United Nations, global achievement of SDGs is only 15%. Further, some studies have revealed significant differences in SDG achievement between countries and regions. In Europe, for example, the analysis by Pizzi et al. [48] of European companies found that only 38.1% of companies in Europe included SDG achievement in their non-financial reports. Analyzing 48 sustainability reports from 20 companies in Greece, Tsalis et al. [49] found that SDG disclosure was low and focused only on reporting on SDG 7, Clean and affordable energy. Analyzing related SDG reports in Italy using 153 companies from eight different industries, Pizzi et al. [12] showed 34% disclosure of SDGs. Gutiérrez-Ponce [50] finds that, on average, listed companies in Spain present a 75% level of information on all SDGs and that significant analogies exist between level of disclosure of GRI-ESG sustainability information and level of performance on the SDGs/ESG.

In analyzing EU real estate companies during the period 2016–2018, Ionaşcu et al. [32] found a gap between planning and realization or achievement of the SDGs. They attributed this gap to the companies' inadequate strategies and knowledge of sustainable engagement. In 2017, Avrampou et al. [51] also analyzed performance on the SDGs in banking in European countries, taking the GRI report as a reference. Their results showed that banking in Europe generally made a low contribution to performance on the SDGs.

In other regions, development of SDGs has also been studied at the industry level. For example, Kumi et al. [11] conducted in-depth interviews with 85 mining and telecommunications companies in Ghana during 2015–2018 and found low levels of SDG achievement. Along the same lines, Erin and Bamigboye [52] found that SDG disclosure in 80 listed companies in Africa was at very low levels. This result was due to lack of management commitment, lack of regulatory compliance, and the potential implications of respondents' business costs. Analyzing the impact of achieving the SDGs on bank profitability in 28 countries, Ozili [44] finds that banks that achieve specific SDGs generally improve their profitability, but different impacts on bank profitability measures are experienced in different regions.

In Southeast Asia, Ike et al. [53] analyzed 16 companies operating in four countries: Vietnam, the Philippines, Thailand, and Indonesia. Their results show that companies prioritize SDGs 4, 8, 9, 11, and 12. In their study of the 100 largest companies in Malaysia, Hamad et al. [54] find an increase in SDG disclosure in Malaysia during 2016–2020, with priority disclosure focusing on SDGs 8, 12, and 13.

In Indonesia, Regulation No. 59 in 2017 became the government's commitment to achieving SDG goals. This regulation regulates how the Indonesian government achieves SDGs at the scale of both regional and central government. The Indonesian Development Plan is based on four pillars aligned with achievement of the SDGs: the economic pillar, the environmental pillar, the social pillar, and the legal certainty pillar [55]. In the same year, Indonesia regulated sustainability reporting for companies, in line with the SDGs.

Several studies specifically analyzed Indonesian companies' contribution to the SDGs. For example, Gunawan et al. [56] analyzed 585 companies in Indonesia using sustainability reporting from 2014 to 2016. Their results showed that Indonesian companies focused most on achieving SDGs 3, 4, 8, 11, and 12. Further, the analysis by Mutiarani and Siswantoro [57] of 34 provinces in Indonesia during 2015–2016 showed that local government's characteristics impacted SDG implementation in Indonesia. Hudaefi's [58] analysis

of 198 fintech (financial technology) companies in Indonesia found that fintech companies have contributed to achievement of SDGs 1, 2, and 3 by contributing financial ideas and innovations to the small and microenterprise (SMEs) sector.

Regarding the Indonesian achievement of the SDGs, 2023 data from the Ministry of National Development Planning of the Republic of Indonesia show fulfillment of 63% of the total 216 indicators of the National Action Plan for SDGs 2021–2024. This figure does not, however, encourage progress in achieving the SDGs at the regional level. Given the high development gap in the current global crisis—especially after the pandemic and the war in Ukraine, which complicates efforts to achieve the SDGs—we lack research that addresses the problem of delineating which factors determine disclosure of the SDGs of companies listed on the Indonesian Stock Exchange, as these companies are obligated to report sustainability.

*Development of Hypotheses*

Many researchers [59–61] have explored the relationship between social responsibility and financial performance. Bonifácio, Neto, and Branco's [62] cross-sector and cross-country analysis obtained evidence of a relationship between corporate social responsibility (CSR) and financial performance. Companies with good financial performance tend to have good relationships with stakeholders through disclosure of non-financial information [12]. In contrast to Gutiérrez-Ponce and Wibowo [16], which shows that profitability has no effect on environmental activities in Indonesian companies, and based on prior theoretical and empirical research, we hypothesize that:

**H1.** *Company profitability is related to SDG disclosure.*

The business industry sensitivity is measured by its level of social, environmental, and ethical commitment. Companies in sectors such as mining, alcohol, or gambling (among others) are classified as not very sensitive [63,64]. Richardson and Welker [65] and Garcia et al. [66] classify companies that produce chemicals, gas, mining, oil, metallurgy, and forestry products as sensitive industries because they are more likely to cause social and environmental damage. Disclosure of sustainability information plays an important role in improving sensitive industries' image and legitimacy [6].

Other studies, such as that of Aqueveque et al. [67], show that controversial industries will further enhance their reputation through their CSR activities. Jo and Na [68] prove that controversial industries must perform more activities related to social and environmental ends to obtain many investors. Singh and Rahman [69] show that implementing SDGs also depends on the industry segment. Based on prior theoretical and empirical research, we pose the following hypothesis:

**H2.** *Companies' sensitivity in environmental matters affects SDG disclosure.*

The size of the board of directors (BOS) plays an important role in corporate governance mechanisms related to SDG disclosure, as does the ability of size to influence corporate strategy [46,70]. Several researchers have studied the impact of BOS on SDG disclosure—for example, Martínez-Ferrero and García-Meca [24] and Husted and Sousa-Filho [71]. They show an increase in positive relations relating the number of director members to CSR practices. Lagasio and Cucari [72] conducted a meta-analysis of 24 articles related to non-financial disclosure. BOS is negatively related to SDG disclosure, however, because it tends to slow decision making [73]. Said et al. [74], who tested quality of corporate governance in Malaysian companies, found that a large BOS results in ineffective communication and leadership and has a negative effect on CSR. Based on prior theoretical and empirical research, we pose the following hypothesis:

**H3.** *BOS is related to SDG disclosure.*

Women have different views from men, relative to personality, leadership style, values they profess, decision-making patterns, etc. [75]. The presence of women on the BOD brings new knowledge on climate change policies, alternative energy, and green building [76]. Gender diversity in the BOD drives environmentally friendly corporate strategies [77]. Isidro and Sobral [78] found that women directors tend to be more supportive of social activities than male directors. Rosati and Faria [23] found that the percentage of women's presence on the BOD had a positive influence on sustainability issues. Further, Zampone et al. [79] show that gender diversity on the BOD positively influences SDG disclosure and that there is a direct relationship between gender diversity on the BOD, SDG disclosure, and the mediating role of the sustainability committee.

Other research shows that female directors on the BOD do not influence sustainability performance [80]. Based on prior theoretical and empirical research, we pose the following hypothesis:

**H4.** *Gender diversity on the BOD is related to dissemination of SDGs.*

Board meetings are needed to develop effective corporate strategies for sustainability performance. Frequency of board meetings is expected to increase transparency and reduce problems of agency. Sekarlangit and Wardhani [80] analyzed performance on SDGs in Southeast Asian companies and found that the holding meetings more frequently increases SDG disclosure. Analyzing companies in Europe, Martínez-Ferrero and García-Meca [24] show that the presence of the board at meetings affects the company's commitment to sustainable development. In their meta-analysis of 24 empirical studies, Lagasio and Cucari [72] demonstrated that the number of meetings increased ESG disclosure. Based on prior theoretical and empirical research, we pose the following hypothesis:

**H5.** *The number of board meetings is related to SDG disclosure.*

## 3. Research Methodology

To achieve our research objectives and answer the questions raised, we conducted an exploratory, descriptive, inferential study. The methods include panel data analysis (using data from each company's website), statistical correlations, and regression models.

The methodology followed is content analysis, defined as follows: "Qualitative content analysis is a research method for the subjective interpretation of the content of text data through the process of systematic classification, coding, and identification of themes or patterns" [81]. This methodology has been widely adopted in studies of corporate disclosure [50,82–85] and is based on the framework for risk communication analysis developed by Beretta and Bozzolan [86]. Previous studies have also used content analysis to measure sustainability performance [6,12,22,76,80,87].

### 3.1. Sample and Data Collection

The study population is companies listed on the Indonesia Stock Exchange (SRI-KEHATI). The SRI-KEHATI Stock Index, first published by the KEHATI Foundation with the Indonesia Stock Exchange (IDX) on 8 June 2009, is a green index that measures the United Nations Principles for Responsible Investment (PRI). The KEHATI Index is currently the only source of investment guidelines that prioritize ESG issues in the Indonesian capital market, with company selection rules using ESG principles and Sustainable Responsible Investment (SRI). The current composition of the SRI-KEHATI Index, which is reviewed and updated twice a year in May and November, comprises 25 stocks of publicly traded companies listed on the IDX.

The study period is 2017–2021, due to ratification in 2017 of the requirements that listed companies present sustainability reports.

We obtained the data from the websites of each company. Firstly, we identified the 25 companies in the Indonesia Selective Index (listed in the SRI-KEHATI Index) in September 2023 by their company name and tax identification number (NIF). Secondly, we classified the companies by sector, following the criteria established by the selective index itself.

To build the database, we first downloaded the annual reports (financial and sustainability) of each of the 25 listed companies and captured the information from each sustainability report by filtering the phrases and words related to the 17 SDGs and their 169 goals. This classification was performed using "RapidMiner" software (https://rapidminer.com/get-started/, accessed on 13 December 2023). A coding procedure was also established to capture information on the SDGs by assigning a value of 1 if the report provided information and 0 otherwise. To measure SDG performance for each goal, this study uses the SDG Compass. Compiled by the UN Global Compact, GRI, and the World Business Council for Sustainable Development (WBCSD), the SDG Compass provides guidance on how companies strategize, manage, and measure the company's contribution to achieving the SDGs.

Additionally, following previous research [2,64–66], the sectors were classified by environmental impact. To collect information on the "environmental sensitivity of the industry", we used a dummy variable, assigning a score of 1 if the company had an impact on the environment and 0 otherwise. All data were transferred to an Excel spreadsheet for processing and study.

Table 1 shows the number and percentage of companies analyzed and classified by sector in the period 2017–2021. The total number of companies that presented sustainability reports in the 5 years analyzed is 110, of which the banking/financial sector represents 22%. This figure is followed by the real estate and construction sector, and then the infrastructure, public services, and transportation sector, each with 15%.

**Table 1.** Number of companies by sector (2017 and 2021).

| Sector | N | % |
|---|---|---|
| Banking/Finance | 24 | 22 |
| Agriculture | 8 | 7 |
| Property and Construction | 17 | 15 |
| Trade and Investment Services | 7 | 6 |
| Various Industries | 6 | 5 |
| Infrastructure, Utilities, and Transport | 16 | 15 |
| Consumer Goods Industry | 13 | 12 |
| Basic and Chemical Industry | 14 | 13 |
| Mining | 5 | 5 |
| Total observation | 110 | 100 |

### 3.2. Variable Measurement

This study uses company characteristics and level of governance as independent variables, as defined in Table 2. Financial profitability (ROE) is defined as a characteristic of companies, in line with previous research by [12,88–90]. Following [2,12], we use industry sensitivity as a variable to classify industries that have a greater impact on the environment relative to those that have less impact. To measure governance level, we use the BOS variable with the number of board members, in line with several previous studies [76,91–93]. The variable gender diversity in the BOD measures percentage of women in the BOD, again aligning with some previous research [76,94,95]. The number of BOD meetings is also used as an independent variable, in line with some previous research [24,72,93].

**Table 2.** Explanation of variables.

| Variables | Labels | Formula |
|---|---|---|
| **Independent Variables** | | |
| Profitability | PROFIT | Net income after taxes divided by average total assets at end of year |
| Environmental Sensitivity of the Industry | SENSITIV | Using dummy variables, the value 1 if includes industries that impact the environment and 0 otherwise |
| Board Size | BOS | Number of board members |
| Board Diversity | BOD | Percentage of women on board |
| Number of Meetings | MEETING | Number of board meetings held in a year |
| **Dependent Variables** | | |
| SDG Disclosure | SDGs | Total disclosure of SDGs divided by Target 17 SDGs |
| **Control variables** | | |
| Size | SIZE | Natural logarithm of total assets at end of year |
| Leverage | LEV | Total Leverage Formula = Total Debt/Shareholder's Equity at end of year |

The dependent variable is the level of SDG disclosure, calculated as total SDG disclosure divided by the 17 SDG targets. Data were obtained from financial and sustainability reports published on the companies' websites.

We use two control variables: size (SZ) and leverage (LEV). SZ is measured using the natural logarithm of total assets and LEV using the company's total debt. These control variables have been used in previous studies, such as [16,24,88,89,96–99]. All financial data for this study are expressed in Indonesian currency, the Rupiah (Rp).

*3.3. Empirical Model*

This study uses panel data that combine time series and cross-sectional data. Before performing the regression test, we tested the classic assumptions to check for normality of the data. Before the regression test, we performed correlation tests between variables measured using Spearman's correlation coefficient. Next, we performed the corresponding tests for normality of the data. First, we tested normality of the data using the Kolmogorov–Smirnov test. Data are distributed normally if the significance value is >0.05. Secondly, we tested multicollinearity to determine whether the relationship between the independent variables was linear. Too high a VIF value is a sign of multicollinearity. Third, we tested for autocorrelation to determine whether a correlation exists between consecutive values in a time series or data series. The Durbin Watson test was used to analyze autocorrelation problems. Fourth, we tested heteroskedasticity using the Breusch–Pagan test. A significance value of >0.05 indicates no symptoms of heteroskedasticity.

This study used multiple regression tests to determine the influence and relationship of all variables on SDG disclosure. This approach coincides with that of several previous studies [54,66,92]. We tested whether the fixed effects model (FEM) or the random effects model was more suitable using the Hausman test. The FEM model is accepted when the probability value is <5%.

Based on the studies cited above, this analysis uses econometric equations with the following multiple regression model:

$$SDGPerf_{i,t} = \alpha + \beta1 PROFIT_{i,t} + \beta2\ SENSITIV_{i,t} + \beta3 BOS_{i,t} + \beta4 BOD_{i,t} + \beta5 MEETING_{i,t} + \beta6 SIZE_{i,t} + \beta7 LEV_{i,t} + \varepsilon_{i,t} \quad (1)$$

where SDGPerf is SDGs disclosure, PROFIT is company profitability, SENSITIV is an industry that impacts the environment, BOS is number of board members in the company,

BOD is proportion of women in the company's board of directors, MEETING is number of board meetings in a year, SIZE is the size of the company's assets, LEV is the company's total debt, "α" is a constant, "β1-7" is the independent variable and the company's control variable, "ε" is an error, "i" is a company, and "t" is a period.

## 4. Results and Discussion

### 4.1. Descriptive Statistics for All Variables

This section is used to answer RQ 1 and RQ 2, on the level of SDG disclosure in Indonesia and the most disclosure by Indonesian companies that contribute to achieving the SDGs.

Table 3 shows that the average ROE (profit) is 18% annually. The variable that measures companies' environmental sensitivity (SENSITIV) indicates that, on average, 79% of companies fall into the category of companies that positively impact the environment, have an average of seven members on their boards of directors, and have 11% women as board members. Furthermore, Indonesian companies hold an average of 42 meetings per year, their total assets average IDR 275,166,373 (in thousands) in an accounting period, and they have an average debt ratio of 75%.

**Table 3.** Descriptive statistics of the variables expressed in percentages (%).

| Variables | Mean | Min | Max |
|---|---|---|---|
| PROFIT | 18 | −22.90 | 146.60 |
| SENSITIV | 79 | 0.00 | 100 |
| BOS | 7.65 | 4 | 12 |
| BOD | 11 | 0.00 | 60 |
| MEETING | 42.23 | 12 | 281 |
| SIZE | 275,166,373 | 3,529,557 | 1,725,611,128 |
| LEV | 75.87 | −16.50 | 348.40 |

Figure 1 presents the level of SDG disclosure in the years analyzed. The results show that SDG disclosure in sustainability reports has been increasing each year. In 2017, 32% of Indonesian companies disclosed the SDGs in their sustainability reports. This level of information increased to 60% in 2021. The trend continues as stakeholders gain awareness and demand information. Sekarlangit and Wardhani [80] showed that companies in Southeast Asia experienced an increase in SDG disclosure each year, while Bose and Khan [100] show that developing countries (including Indonesia) have rapidly increased their SDG disclosure.

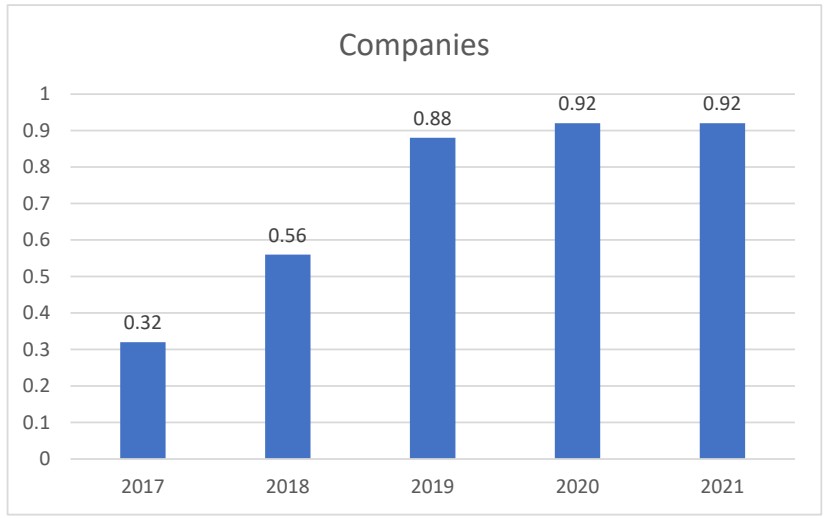

**Figure 1.** Companies that disclose SDGs every year.

To answer RQ2, Figure 2 shows the priority in disseminating the SDGs—that is, which SDGs are most widely disseminated in Indonesia.

**Figure 2.** SDG disclosure priorities in Indonesia (2017–2021).

The highest levels of disclosure are in SDG 3 (Health and well-being) and SDG 4 (Quality of education), each with a value of 0.68. These results indicate that the government's development policies focus on qualification of its human resources through a commitment to education and health to impact social well-being and improve economic development.

Another interesting result is the commitment to preservation of the environment and prevention of climate damage. Several SDG targets that have a disclosure value greater than >0.50—for example, SDG 6 (Clean water and sanitation), SDG 7 (Affordable and clean energy), SDG 9 (Industry, innovation, and infrastructure), SDG 13 (Climate action), and SDG 15 (Life on land).

However, SDG 14 (Life below water) has among the lowest disclosure values in Indonesia. This result shows that Indonesian companies still have little concern for river and marine ecosystems. Fifty percent of the population lives in coastal areas, and a healthy and protected ocean is crucial to the country's development and prosperity. This finding is in accordance with research [32] showing that most companies determine their SDG priorities according to their company's strategy and activities.

Given this problem with SDG 14, an agreement was signed in November 2022 between the government and 28 organizations—eight government ministries, eight UN entities, and 12 partners to join forces within the framework of the National Agenda Action Associations Blue (National Blue Agenda Actions Partnership).

Our analysis of the dissemination of other SDGs related to the environment shows that only SDG 13 reaches prioritization levels over 50%. However, environmental onsite sewage disposal systems (OSDs) that aim to achieve more sustainable cities (SDG 11), and responsible consumption and production (SDG 12), as well as signing of alliances to achieve the objectives (SDG 17), obtain lower levels of disclosure or prioritization.

Social objectives—such as SDG 5 on gender equality, SDG 1 on ending poverty, and SDG 16 on promoting just, peaceful, and inclusive societies—also fail to reach 50% in the prioritization and dissemination levels.

### 4.2. Correlation Result

This test is used to determine the direction of the relationships between all variables. We use the Spearman correlation test to determine the direction of the relationship between the independent and dependent variable.

Table 4 shows the results of the correlation test between all variables. We observe that SDGs have a positive correlation with sensitive industries (0.359). This finding shows that industries that are highly sensitive to the environment (such as mining, oil, and gas) commit increasingly to environmental issues to improve the company's image [67].

**Table 4.** Correlation test for all variables.

|  | SDGs | Profit | Ind Sens | BOS | BOD | Meeting | Size | Lev |
|---|---|---|---|---|---|---|---|---|
| SDGs | 1 | −0.059 * | 0.359 ** | 0.315 ** | 0.281 ** | 0.204 * | 0.365 ** | −0.047 |
| Profit |  | 1 | 0.089 | 0.158 | 0.565 ** | −0.173 | −0.091 | −0.159 |
| Sensitiv. |  |  | 1 | −0.754 ** | −0.208 * | −0.474 ** | −0.697 ** | −0.086 |
| BOS |  |  |  | 1 | 0.449 ** | 0.195 * | 0.603 ** | −0.099 |
| BOD |  |  |  |  | 1 | 0.046 | 0.193 * | 0.009 |
| Meeting |  |  |  |  |  | 1 | 0.238 * | 0.503 * |
| Size |  |  |  |  |  |  | 1 | −0.024 |
| Lev |  |  |  |  |  |  |  | 1 |

Note: Correlation is significant at * 0.05 ** 0.01.

SDGs have a positive relationship to BOS (0.315). Thus, the number of members on the BOD aligns with the company's CSR performance [71]. SDGs also have a positive correlation to board diversity (0.281), showing that the presence of women on the BOD encourages companies to behave in an environmentally friendly manner [78]. Further, SDGs have a positive correlation with number of meetings (0.204), in line with Martínez-Ferrero and García-Meca [24], who found that the presence of the board at the meeting would increase the company's commitment to sustainable development.

Companies' financial profitability is, however, inversely related to SDG disclosure. This result may be due to the fact that sustainability activities use a large amount of financial and organizational resources in the short-term, reducing profits [45,89,101–103].

### 4.3. Regression Results to Determine Factors of SDG Disclosure

This section presents the results to answer RQ3 and contrasts the hypotheses related to the determining factors in listed companies' SDG disclosure. Table 5 shows the results of the normality tests (multicollinearity, autocorrelation, and heteroskedasticity). The normality test shows that the data are normally distributed, as seen from the Kolmogorov–Smirnov significance value of 0.138 > 0.05. The tolerance and VIF values for all variables are >0.1, and the VIF values <0.10 indicate no symptoms of multicollinearity. The heteroskedasticity test shows the significance values for all variables is >0.05, indicating no symptoms of heteroskedasticity. The Durbin Watson value of 1.875 is within the range of 1.8 and 2.17 [104], indicating no symptoms of autocorrelation.

Before regression testing, we tested which model, FEM or REM, was more suitable. Since the Hausman test results showed a probability value of 0.0030 < 5%, the FEM was accepted. Table 6 shows the results of the proposed regression model to determine which factors influence SDG diffusion in Indonesia. The hypothesis is accepted if the significance value is less than 5%.

**Table 5.** Normality tests for data quality.

| Model | Heteroskedasticity Test | Collinearity Test | |
|---|---|---|---|
| | | Tolerance | VIF |
| PROFIT | 0.693 | 0.598 | 1.673 |
| SENSITIV | 0.085 | 0.120 | 8.325 |
| BOS | 0.809 | 0.249 | 4.013 |
| BOD | 0.522 | 0.521 | 1.920 |
| MEETING | 0.071 | 0.448 | 2.231 |
| SIZE | 0.062 | 0.121 | 8.282 |
| LEV | 0.837 | 0.685 | 1.460 |
| Durbin Watson | | 1.875 | |
| Kolmogorov–Smirnov Z | | 1.156 | |
| Asymp. Sig (2-tailed) | | 0.138 | |

**Table 6.** Regression results.

| Model | Coeff (Sig) |
|---|---|
| Independent Variables: | |
| PROFIT | 0.000 (0.785) |
| SENSITIV | 0.061 (0.778) |
| BOS | −0.030 (0.247) |
| WOMEN | 0.788 (0.013) * |
| MEETING | 0.002 (0.077) ** |
| Control Variables: | |
| SIZE | 3.706 (0.064) |
| LEV | −0.001 (0.103) |
| Adj. R Square | 0.168 |
| F Test (Sig.) | 0.000 |
| Hausman test (Prob.) | 0.0030 * |
| Fixed Effect Model (FEM) | Yes |

Note: significant at * 0.05 ** 0.10.

The regression test results show that H1 is rejected—that is, that profit has no effect on SDG disclosure, as seen from the significance value of $0.785 > 5\%$. This result does not agree with that of [12], which states that profitability influences SDG disclosure. Our results show that companies' sensitivity in environmental matters does not affect disclosure of SDGs. We thus reject H2 with a significance value of $0.778 < 5\%$, indicating that the sensitivity of the industry will not affect disclosure of the SDGs. SDG disclosure thus does not depend on the type of industry (sensitive or non-sensitive). In fact, the existence of SDG reporting may be an alternative means to address industry controversy to reduce stakeholder pressure and social supervision and to improve the company's image [6].

The results of this study show that the BOS is not related to SDG disclosure (H3), since the significance value of $0.247 > 5\%$ for BOS does not affect that disclosure. This finding coincides with Giannarakis [92], who shows that the BOS does not play a significant role in CSR disclosure, possibly because the board only contributes at the policy level, not the implementation level.

H4 is accepted, as seen from the significance value of $0.0013 < 5\%$. The presence of women on the BOD is thus significantly positive for SDG disclosure. This finding aligns with the results of Rosati and Faria [23], who show that women provide a better perspective and support environmental activities. It also shows that the presence of women on the BOD will increase control over the board and be favored by investors, reducing agency costs [105].

The results for H5 indicate that number of board meetings is related to disclosure of SDGs, based on the significance value of $0.077 < 10\%$. This finding coincides with the

findings of Sekarlangit and Wardhani [80], who show that the number of meetings provides good knowledge of how committed companies are to SDG disclosure.

## 5. Conclusions, Implications, and Research Limitations

This study assesses implementation of SDGs in Indonesia and the factors that influence this implementation. Using data on Indonesian companies published on the Indonesia Stock Exchange in the period 2017–2021, we tested whether company characteristics (profitability and industry sensitivity) and governance level (board of size, board diversity, and number of meetings) impacted SDG disclosure in Indonesia. We also used control variables such as company size and debt (debt-to-equity ratio).

The findings, first, prove that disclosure of SDGs in Indonesia has increased every year. Whereas in 2017 only 32% of companies disclosed SDGs, this figure increased to 92% in 2021. Indonesia is thus seriously committed to achieving the 2030 SDG target. Second, companies in Indonesia focus on disclosing SDGs in social and environmental areas such as SDGs 6 (Clean energy and sanitation), 7 (Affordable and clean energy), 8 (Decent work and economic growth), 9 (Industry innovation and infrastructure), 13 (Climate change), and 15 (Life on land). The highest disclosures are in SDG 3 (Good health and well-being) and 4 (Quality education). The lowest is in SDG 14 (Life below water).

These results can be explained by Indonesia's status as a developing country. Both the government and companies develop strategies to improve people's well-being through a commitment to education and health to impact social well-being and improve economic development.

Although many of the companies analyzed are aware that their activities affect the environment, the environmental SDGs are not among their disclosure priorities. This finding may indicate that the Indonesian companies analyzed are trying to comply with the 2017 regulatory obligation to submit an environmental report that gives them a green or ecological image but are not as concerned about actual sustainability performance. Furthermore, it has been observed that the social SDGs have much to improve in terms of dissemination and performance. It is true that it is important for developing countries to create an environmental organizational culture that considers the costs of ecological transformation and sustainability of long-term investments. Since this path is longer in developing countries, it is important that they advance along that path.

The previous conclusion is reinforced by the finding that the companies' financial profitability is inversely related to SDG disclosure. This inverse relationship can be explained by the fact that short-term sustainability activities use a large amount of financial and organizational resources, diminishing the benefits. We also find that ROE (profit) does not influence the SDG disclosure strategy in the Indonesian companies analyzed.

Although companies' sensitivity in environmental matters does not affect Indonesian companies' decision to report and disclose the SDGs, gender diversity (or the presence of women on the BOD) and the number of board meetings are positively related to SDG disclosure strategy.

This finding accords with several previous studies that argue that women have a more innovative perspective on environmental conditions and climate change policies [76]. The number of meetings also positively impacts the environment. It is likely that meetings conducted by the board lead to more pro-sustainability alignment and decision making.

In conclusion, we note that the companies analyzed perform some greenwashing, orienting their marketing image to ecological positioning, while their actions less clearly or specifically favor the environment. The reports presented on "green communication" thus do not always mean that companies are more respectful of the environment or that they have acquired a commitment to the environment. It is important, however, that progress continues in a commitment to report on SDGs and ESG issues by measuring them with GRI indicators or the European sustainability reporting standards (ESRS) indicators.

This study has significant implications for stakeholders, governance policy makers, and academia. For stakeholders, it clarifies the relationship between SDG disclosure and

financial and governance factors of listed Indonesian companies. Implications for practitioners can provide input on what factors will improve SDG disclosure. For investors, the findings relate financial performance to the characteristics of companies' BOD. Further, the results help policy makers to understand how and why companies change their reporting and transparency practices, affecting the credibility and effectiveness of corporate and sustainability reporting. For decision makers such as the government, our study provides an overview of Indonesian companies' contribution to the achievement of SDGs to help the government develop the right SDG-related policies. For academics, the study contributes to an emerging body of literature aligned with sustainability reporting, corporate governance, and non-financial information in an understudied emerging markets context, and therefore addresses development and governance gaps that can explain the problems involving information on and development of the SDGs.

This study has several limitations. First, the sample is only companies in the SRI-KEHATI Index, namely 25 companies with good performance relative to the environment. For better results, one could use a larger sample, such as all companies listed on the Indonesia Stock Exchange.

**Author Contributions:** Conceptualization, H.G.-P.; methodology, H.G.-P. and S.A.W.; software, S.A.W.; validation, H.G.-P. and S.A.W.; formal analysis, H.G.-P.; investigation, writing—original draft preparation, H.G.-P.; writing—review and editing, H.G.-P. All authors have read and agreed to the published version of the manuscript.

**Funding:** This research received no external funding.

**Institutional Review Board Statement:** Not applicable.

**Informed Consent Statement:** Not applicable.

**Data Availability Statement:** Data are contained within the article.

**Conflicts of Interest:** The authors declare no conflict of interest.

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
