# Peer review of "Sustainability Reports and Disclosure of the Sustainable Development Goals (SDGs): Evidence from Indonesian Listed Companies"

_sustainability, doi:10.3390/su152416919_

Round 1

Reviewer 1 Report

Comments and Suggestions for Authors

The paper investigates the factors that are likely to induce a listed company to report on their supposed contributions to the UN SDGs in Indonesia. There are certainly some generally useful insights such as the increase in number of companies in the country that do SDG report and to which of the SDGs they pay particular attention (and which one they neglect). However the main empirical part relies on qualitative data published on company websites that is not beyond doubts because what is reported by a company is often highly selective when it comes to the contribution to SDGs (a lot is merely repackaged as it is based on earlier activities).

The paper then compares the research results to prior research papers that tested similar hypotheses using similar available data in other countries. The authors suggest that prior findings are largely confirmed by their findings. But are they really comparable and what about the odd result that profitability correlates negatively with SDG reporting? The main argument was so far that there is a win-win potential even on the short-run since companies that are able to present a sustainability certificate in tenders have higher chances to win. Maybe having an SDG report (purely based on self-assessment) in addition to a more quantitative sustainability certificate (e.g. ecovadis, ISO) is not really an added value. This makes sense because the qualitative nature of SDG evaluation tools such as SDG compass are increasingly confronted with skepticism (Greenwashing suspicion). The paper does not address this problem that has gained in importance in 2023 in view of several scandals and serveral new regulatory initiatives in the EU and the US to reduce greenwashing. In addition, it remains unclear for the reader what the concrete message of this paper is. Should more Indonesian companies now report on more SDGs? But what if it largely remains rhetoric.

Overall I recommend to the authors

- to give their paper a more clear direction (focusing on one particular hypothesis and finding that could be of general interest rather than reporting on everything without embedding it into the larger discourse)

- to write a word of caution regarding the data quality

- to discuss also the currently criticized fact that the mere reporting makes companies look sustainable even though it does not reveal much about their actual sustainability performance compared to their peers.

Comments on the Quality of English Language

Many typos and convoluted sentences. I would ask a native english speaker to proofread the paper

Author Response

Comments and Suggestions for Authors

The paper investigates the factors that are likely to induce a listed company to report on their supposed contributions to the UN SDGs in Indonesia. There are certainly some generally useful insights such as the increase in number of companies in the country that do SDG report and to which of the SDGs they pay particular attention (and which one they neglect). However the main empirical part relies on qualitative data published on company websites that is not beyond doubts because what is reported by a company is often highly selective when it comes to the contribution to SDGs (a lot is merely repackaged as it is based on earlier activities).

ANSWER

Thanks for your comments.

Indeed, the data was obtained from the sustainability reports published on the websites of the 25 companies listed on the Indonesian stock exchange (SRI-KEHATI). The chosen study period is 2017-2021 due to the entry into force, in 2017, of the obligation to present sustainability reports for listed companies.

Unfortunately, there is no possibility of obtaining a database with sustainability indicators or compliance with the SDGs. Therefore, the database for the investigation must be built using research techniques verified by previous studies and which we have detailed in our manuscript.

To build the database, we first downloaded the annual reports (financial and sustainability) of each of the 25 listed companies and captured the information from each sustainability report by filtering the phrases and words related to the 17 SDGs and their 169 goals. This classification was carried out using the “RapidMiner” software (https://rapidminer.com/get-started/).

Also, to measure SDG performance for each goal, this research uses the SDG compass. The SDG Compass compiled by the UN Global Compact, GRI and the World Business Council for Sustainable Development (WBCSD) provides guidance on how companies strategize, manage, and measure the company's contribution to achieving the SDGs.

We agree with the reviewer that qualitative information can have a lot of marketing (Greenwashing suspicion) but if the obligation to present specific sustainability indicators or standards is not regulated, there may always be that doubt.

The paper then compares the research results to prior research papers that tested similar hypotheses using similar available data in other countries. The authors suggest that prior findings are largely confirmed by their findings. But are they really comparable and what about the odd result that profitability correlates negatively with SDG reporting?

ANSWER

Thanks for the observation.

We have now made these matters clearer. We have clarified the problems and characteristics of a developing country like Indonesia and the difficulties in investing in environmental, social and governance issues.

The finding that profitability is negatively correlated with SDG reporting has also been clarified. This result may be since in the short-term sustainability activities will use a large amount of financial and organizational resources, which will reduce profits [45,89,101,102,103].

The main argument was so far that there is a win-win potential even on the short-run since companies that are able to present a sustainability certificate in tenders have higher chances to win. Maybe having an SDG report (purely based on self-assessment) in addition to a more quantitative sustainability certificate (e.g. ecovadis, ISO) is not really an added value. This makes sense because the qualitative nature of SDG evaluation tools such as SDG compass are increasingly confronted with skepticism (Greenwashing suspicion). The paper does not address this problem that has gained in importance in 2023 in view of several scandals and serveral new regulatory initiatives in the EU and the US to reduce greenwashing. In addition, it remains unclear for the reader what the concrete message of this paper is. Should more Indonesian companies now report on more SDGs? But what if it largely remains rhetoric.

ANSWER

Thanks for the observation.

This matter has been addressed in the conclusions. There has also been discussion about the path taken by companies listed on the Indonesian stock exchange and compliance with the obligation, as of 2017, to present sustainability reports. Which does not mean that there is not a certain greenwashing in these reports by orienting their marketing image towards an ecological positioning while their actions are not so clear or are not specified in favor of the environment.

Overall I recommend to the authors

- to give their paper a more clear direction (focusing on one particular hypothesis and finding that could be of general interest rather than reporting on everything without embedding it into the larger discourse)

ANSWER

Thanks for the observation.

Now the namuscript has been completely reviewed and a clear orientation of the problem, objectives, results and conclusions has been given.

- to write a word of caution regarding the data quality

Thank you,

Limitations have been addressed.

- to discuss also the currently criticized fact that the mere reporting makes companies look sustainable even though it does not reveal much about their actual sustainability performance compared to their peers.

ANSWER

Thanks for the observation.

This matter has now been discussed in the results and conclusions.

 Comments on the Quality of English Language

Man, typos and convoluted sentences. I would ask a native english speaker to proofread the paper.

ANSWER

Thanks for the observation.

The manuscript has now been fully reviewed and corrected by a native English language reviewer

Reviewer 2 Report

Comments and Suggestions for Authors

*Review Report

*Paper Title: Sustainability reports and disclosure of the Sustainable Development Goals (SDGs): Determinants of the disclosure of SDGs in listed companies in Indonesia

*Abstract:

This review critically evaluates the paper titled "Sustainability reports and disclosure of the Sustainable Development Goals (SDGs): Determinants of the disclosure of SDGs in listed companies in Indonesia " from a critical perspective. The following observations are provided to elevate the scholarly quality and rigor of the research:

*1) Strengthening the Introduction:

The introduction necessitates significant refinement to construct a more cohesive and engaging narrative. The current version lacks the requisite fluidity and coherence expected in academic publications of this caliber. A revised introduction should establish a compelling and logically structured foundation for the research.

*2) Enhancing Section Connectivity:

The transitions between sections require substantial improvement to ensure a seamless and logically coherent paper structure. At present, the transitions are somewhat disjointed, detracting from the overall readability. Improved section connectivity is essential.

*3) Addressing Citation-Reference Discrepancies:

It is imperative to resolve the discrepancies between in-text citations and the references section. Ensuring that all cited sources are meticulously included in the reference list is fundamental for maintaining scholarly accuracy and integrity.

*4) Engagement of Professional Proofreading and Editing:

The paper is marred by numerous grammatical errors and typographical flaws, which are detrimental to the quality of an academic publication. It is strongly recommended to engage professional proofreading and editing services to rectify these issues and align the paper with the rigorous standards expected in academic research.

*5) Incorporation of Relevant Citations:

To enhance the academic rigor and alignment of the paper with the standards of top-tier academic publications, it is advisable to integrate supplementary citations. One such citation, among others, could be:

Erin, O. A., & Bamigboye, O. A. (2021). Evaluation and analysis of SDG reporting: Evidence from Africa. Journal of Accounting & Organizational Change, 18(3), 369-396.

Zampone, G., Nicolò, G., Sannino, G., & De Iorio, S. (2022). Gender diversity and SDG disclosure: the mediating role of the sustainability committee. Journal of Applied Accounting Research.

*6) Adapting Language for a Multidisciplinary Audience:

Recognizing the diverse and multidisciplinary readership that sustainability papers attract, it is essential to make the language as inclusive and accessible as possible. This adaptation ensures that the paper caters to a broader spectrum of readers.

*7) Overall comments:

The paper offers a comprehensive investigation into the determinants of Sustainable Development Goals (SDGs) disclosure among companies listed on the Indonesian stock exchange between 2017 and 2021. Employing an exploratory study methodology, including panel data analysis and regression models, the research unveils a substantial 60% increase in SDG disclosure within sustainability reports during this period. Notably, SDG 3 (Health and well-being) and SDG 4 (Quality education) exhibit the highest disclosure levels, while SDG 14 (Life below water) displays the lowest disclosure. Empirical insights confirm that governance factors, specifically the presence of women on the Board of Directors (BOD) and the number of board meetings, exert a positive influence on SDG disclosure among listed Indonesian companies. In contrast, factors related to profitability, environmental sensitivity, and board size do not significantly impact SDG disclosure, consistent with prior studies emphasizing the unique perspective of women on environmental matters and climate policies.

The paper holds significant implications for a broad range of stakeholders, encompassing academics, practitioners, policymakers, and governments striving to attain the 2030 SDG agenda. The research sheds light on the intricate relationship between financial, governance, and SDG disclosure factors for Indonesian companies. These insights provide valuable guidance to investors, policymakers, and practitioners for enhancing SDG disclosure.

The paper underscores certain limitations, primarily stemming from the limited availability of SDG data within sustainability reports, especially within the Indonesian stock exchange. The research sample is confined to the SRIKEHATI index, encompassing only 25 companies. Future research endeavors may broaden the sample size and introduce additional variables for a more comprehensive analysis.

In conclusion, the paper exhibits significant potential but requires substantial refinement to adhere to the rigorous standards expected of top-tier academic publications. Adhering to these recommendations will unequivocally elevate the paper's overall quality.

The author is urged to diligently consider these recommendations and conduct comprehensive revisions before resubmission.

Yours sincerely,

Comments on the Quality of English Language

The paper is marred by numerous grammatical errors and typographical flaws, which are detrimental to the quality of an academic publication. It is strongly recommended to engage professional proofreading and editing services to rectify these issues and align the paper with the rigorous standards expected in academic research.

Author Response

Comments and Suggestions for Authors

This review critically evaluates the paper titled "Sustainability reports and disclosure of the Sustainable Development Goals (SDGs): Determinants of the disclosure of SDGs in listed companies in Indonesia " from a critical perspective. The following observations are provided to elevate the scholarly quality and rigor of the research:

 *1) Strengthening the Introduction:

The introduction necessitates significant refinement to construct a more cohesive and engaging narrative. The current version lacks the requisite fluidity and coherence expected in academic publications of this caliber. A revised introduction should establish a compelling and logically structured foundation for the research.

ANSWER

Thank you very much for all the comments.

The introduction has been completely revised and restructured to make it more fluid and lays the foundation for the research.

 *2) Enhancing Section Connectivity:

The transitions between sections require substantial improvement to ensure a seamless and logically coherent paper structure. At present, the transitions are somewhat disjointed, detracting from the overall readability. Improved section connectivity is essential.

ANSWER

All sections have been reviewed and improved. to improve the fluidity of the entire document.

 *3) Addressing Citation-Reference Discrepancies:

It is imperative to resolve the discrepancies between in-text citations and the references section. Ensuring that all cited sources are meticulously included in the reference list is fundamental for maintaining scholarly accuracy and integrity.

ANSWER

Thanks for the observation.

All cited sources and references have been carefully checked for errors and inaccuracies.

 *4) Engagement of Professional Proofreading and Editing:

The paper is marred by numerous grammatical errors and typographical flaws, which are detrimental to the quality of an academic publication. It is strongly recommended to engage professional proofreading and editing services to rectify these issues and align the paper with the rigorous standards expected in academic research.

ANSWER

Thanks for the observation.

The manuscript has now been fully reviewed and corrected by a native English language reviewer.

*5) Incorporation of Relevant Citations:

To enhance the academic rigor and alignment of the paper with the standards of top-tier academic publications, it is advisable to integrate supplementary citations. One such citation, among others, could be:

Erin, O.A. and Bamigboye, O.A. (2022).Evaluation and analysis of SDG reporting: evidence from Africa. Journal of Accounting & Organizational Change, 18 No. 3, pp. 369-396. https://doi.org/10.1108/JAOC-02-2020-0025

Zampone, G., Nicolò, G., Sannino, G. and De Iorio, S. (2022). Gender diversity and SDG disclosure: the mediating role of the sustainability committee. Journal of Applied Accounting Research, Vol. ahead-of-print No. ahead-of-print. https://doi.org/10.1108/JAAR-06-2022-0151

Answer.

Thanks for the references.

They are now included in the manuscript.

*6) Adapting Language for a Multidisciplinary Audience:

Recognizing the diverse and multidisciplinary readership that sustainability papers attract, it is essential to make the language as inclusive and accessible as possible. This adaptation ensures that the paper caters to a broader spectrum of readers.

ANSWER

Thanks for the observation.

The manuscript has been completely revised to ensure inclusive and accessible language

*7) Overall comments:

ANSWER

Our sincere thanks for reading our manuscript and for all your insightful observations.

 Comments on the Quality of English Language

The paper is marred by numerous grammatical errors and typographical flaws, which are detrimental to the quality of an academic publication. It is strongly recommended to engage professional proofreading and editing services to rectify these issues and align the paper with the rigorous standards expected in academic research.

ANSWER

The manuscript has now been fully reviewed and corrected by a native English language reviewer.

Reviewer 3 Report

Comments and Suggestions for Authors

See the attachment.

Author Response

Structuring and Exposition

  1. The article title could be shortened. Currently, some words, such as disclosure or sustainability, appear twice in the title. Perhaps the second clause could be shortened to just “Evidence from Indonesia” or something similar.

Answer:

Thanks for the suggestion.

The title has now been reformulated. Sustainability reports and disclosure of the Sustainable Development Goals (SDGs): Evidence from Indonesian listed companies.

2. The abstract could also be shortened slightly to make it sharper and more focused. For example, the last two sentences seem redundant.

Answer

Thanks for your suggestion. The last sentence has been removed.

 3. The introduction would benefit from a moderate restructuring. The first motivation is a bit too long. On the other hand, the findings are presented and discussed very succinctly. The authors could bring them forward and elaborate on them. Ideally, the authors should start explaining what they are doing in the 2nd or 3rd paragraph and start reporting the results in the 3rd or 4th paragraph. The outline of the paper should follow. Authors can refer to John Cochrane's writing tips, which I personally found very helpful (https://www.fma.org/assets/docs/membercontent/writing_cochrane.pdf).

Answer

Thanks for the observations and for the suggestion and reference on writing tips from John Cochrane. Now the introduction has been restructured and its common thread has been improved.

Theoretical Basis

  1. The literature review is well-structured and comprehensive. However, the authors could also link their paper to the interplay between SDGs and firm performance (Khaled et al., 2021; 2022, Ozili, 2023, to name just a few).

Answer

Thanks for the references.

Both have been included and we found them interesting.

References

Khaled, R., Ali, H., & Mohamed, E. K. (2021). The Sustainable Development Goals and corporate sustainability performance: Mapping, extent and determinants. Journal of Cleaner Production, 311, 127599. https://doi.org/10.1016/j.jclepro.2021.127599

Ozili, P. K. (2023). Sustainable Development Goals and bank profitability: International evidence. Modern Finance, 1(1), 70-92. https://doi.org/10.61351/mf.v1i1.44

Data and Methods

  1. The authors use only firms available at the end of our sample. This approach may introduce a survivorship bias. What about the firms that existed in 2017 and 2018 but were delisted? This issue should be highlighted in the list as a potential limitation of the study.

Answer:

Thanks for the observation.

This and other limitations have been addressed in section 5.

  1. Many readers may not be familiar with the Indonesia Selective Index. It would be good for the authors to explain it in more detail.

Answer:

Thanks for the observation.

We have added information related to the definition of the SRI KEHATI index in section 3.1.

  1. The explanation of the variables in Table 2 should be more detailed. Let us take PROFIT as an example. Over what period is net profit calculated? Is it the last four quarters? Or the last calendar year? What about total assets: is it the beginning of the period, or the end of the period, or some kind of average? The same goes for other measures.

Answer:

Thanks for the observation.

Now the information on the variables in Table 2 has been completed and detailed so that there are no doubts about interpretation.

  1. What currency is the study in? In what currency are all accounting and financial data expressed?

Answer:

Thanks for the observation.

 We have added the currency related information in section 3.2. and in the

presentation of the results.

  1. Authors should ensure that all figures are self-explanatory. Currently, many tables lack sufficient notes to be fully understood without scanning the entire article.

Answer:

We have added explanatory information in each table according to your suggestions.

  1. The authors should clarify the units used in Table 3. Also, I have doubts that their interpretation is correct. For example, on line 363, the authors write "the percentage of women on the board is 0.11%." Does this mean that there is about 1 woman for every 1000 board members?

Answer:

Thanks for the observation and your suggestion.

Table 3 has now been revised and its units have been clarified.

The text on its interpretation has also been completely revised.

  1. The tests reported in section 4.2 are unclear. What kind of correlation is it? Pearson's product moment? Did the authors compute it over the entire panel, or is it a time series average of cross-sectional correlations?

Answer:

Thanks for the observation.

Section 4.2 has been revised and this issue has been clarified. This research uses the Spearman correlation test. The reason for using Spearman is because the data is normally distributed.

  1. What is a "classical test of hypotheses"?

Answer:

Thanks for the observation.

Section 3.2 has been revised and this issue has been clarified.

The classical assumption test is a series of statistical tests used to test a number of basic assumptions that must be met in classical regression analysis and several other statistical methods. These include: normality test, multicollinearity test, autocorrelation test and heteroscedasticity test. These assumptions involve the characteristics of the data and the relationships between variables that form the basis of the statistical model. This explanation is already in section 3.2.

  1. What type of regressions do the authors run? Cross-sectional or panel? If so, what kind of effects do they assume: fixed or random, and why?

Answer:

Thanks for the observation.

Section 3.2 has been revised and this issue has been clarified.

 This research uses panel data which combines cross-sectional data with time series as we have explained in section 3.2. We have added an explanation regarding the selection of a suitable estimation model for this research, between the Fixed Effect Model (FEM) or the Random Effect Model (REM).

Editorial Comments

  1. In the equation in section 2 (shouldn't it be numbered?), it would be better to report i and t as subscripts of the respective variables.

Answer:

Thanks for the observation.

The section has been revised, the equation has been numbered, and we have changed the “i” and “t” to the subscript style.

Round 2

Reviewer 3 Report

Comments and Suggestions for Authors

The authors have properly resolved my comments. I believe the article is acceptable in the current form.